# Characterization of Atmospheric Deposition as the Only Mineral Matter Input to Ombrotrophic Bog

**Valentina Pezdir \*, Martin Gaberšek and Mateja Gosar** 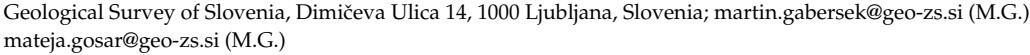

Geological Survey of Slovenia, Dimičeva Ulica 14, 1000 Ljubljana, Slovenia; martin.gabersek@geo-zs.si (M.G.); mateja.gosar@geo-zs.si (M.G.)

\* Correspondence: valentina.pezdir@geo-zs.si

**Abstract:** Ombrotrophic peatlands contain a very small percentage of mineral matter that they receive exclusively from atmospheric deposition. Mineral matter deposited on the Šijec bog was characterized using scanning electron microscopy coupled with energy dispersive spectroscopy (SEM/EDS). We collected solid atmospheric deposition from snow, rainwater, and using passive samplers. Samples were collected at average atmospheric conditions and after two dust events. Size, morphology, and chemical composition of individual particles were determined. We distinguished four main particle groups: silicates, carbonates, organic particles, and Fe-oxyhydroxides. Silicate particles are further divided into quartz and aluminosilicates. Proportions of these groups vary between samples and between sample types. In all samples, silicate particles predominate. Samples affected by dust events are richer in solid particles. This is well observed in passive deposition samples. Carbonates and organic particles represent smaller fractions and are probably of local origin. Iron-oxyhydroxides make up a smaller, but significant part of particles and are, according to their shape and chemical composition, of both geogenic and anthropogenic origin. Estimated quantity and percentage of main groups vary throughout the year and are highly dependent on weather conditions. Dust events represent periods of increased deposition and contribute significantly to mineral matter input to peatlands.

**Keywords:** atmospheric deposition; SEM/EDS; mineral matter; peatland





## 1. Introduction

Peatlands are specific anoxic and acidic environments that have the potential to retain and store large amounts of major and trace elements [1,2]. Ombrotrophic peatlands are limited to the atmosphere as a source of water and mineral matter [3,4], and are believed to be good geochemical archives of air quality over the past centuries [1,5,6]. This is based on the assumption that trace elements are immobile in the peat matrix and that accumulation in the record represents a direct measure of atmospheric deposition [7,8]. However, many authors considered different influences on the peat mineral matter record. Some major and trace elements may be removed from the peatland (e.g., Zn and Ni), while others are retained or moved within the peat profile (e.g., Pb and Cu) [9]. Dissolved organic matter derived from the decomposition of plant remains in peatlands can also act as a major carrier of trace elements from peatlands to streams [10]. Others also studied changes that occur during the transition of peatland from minerotrophic to ombrotrophic peat and how the peatland size affects mineral matter [11]. Additionally, changes in mineral phases in peat were also studied [12,13]. Nevertheless, ombrotrophic peatlands are highly dependent on atmospheric processes and geographic location. In Europe, larger peatland areas are located in northern Europe, Russia, the United Kingdom, and Ireland [14], where higher levels of precipitation are recorded.

Mineral matter is deposited on peatlands as both dry and wet atmospheric deposition and represents about 1% of the total peat mass in ombrotrophic peatlands [15]. Periods of

increased atmospheric deposition, such as dust events, could be one of the most important factors contributing to the total amount of mineral matter in peatlands.

Dust events transport airborne particles over long distances and are the subject of numerous studies of atmosphere, environment, and health [16–18]. Most dust events in Europe originate from North Africa (Sahara Desert) [19,20], with most events occurring in spring due to cyclones [21]. Dust events in Europe can also originate from other areas such as East Asia [20], and are very rarely associated with areas in Australia, South Africa, South America, and the United States [22]. Some authors also observed dust event records within peat [23].

Peatlands in Slovenia are one of the southernmost ombrotrophic peatlands in Europe [24], and some are located on the Pokljuka karst plateau in northeastern Slovenia (Figure 1). The plateau rises at an elevation of 1200 m to 1500 m and has a relatively flat topography. In the south, southwest, and east, the plateau is surrounded by valleys, while in the northwest it borders on a higher mountain range. Due to the karst environment, surface water is found only in karst springs and peatlands [25,26]. Among them, our study area of the Šijec bog is the largest and covers approximately 15 hectares. It is located 1194 m above sea level. The bog formed after the last ice age as four separate basins [27]. Due to its altitude, the Šijec bog as well as the Pokljuka plateau have a colder climate than the Slovenian average. For example, in March 2020, the average temperature in Slovenia was 7.2 °C [28], while the average temperature on the Pokljuka plateau was −1.3 °C (measured at the Rudno polje meteorological station, 5 km from the Šijec bog).

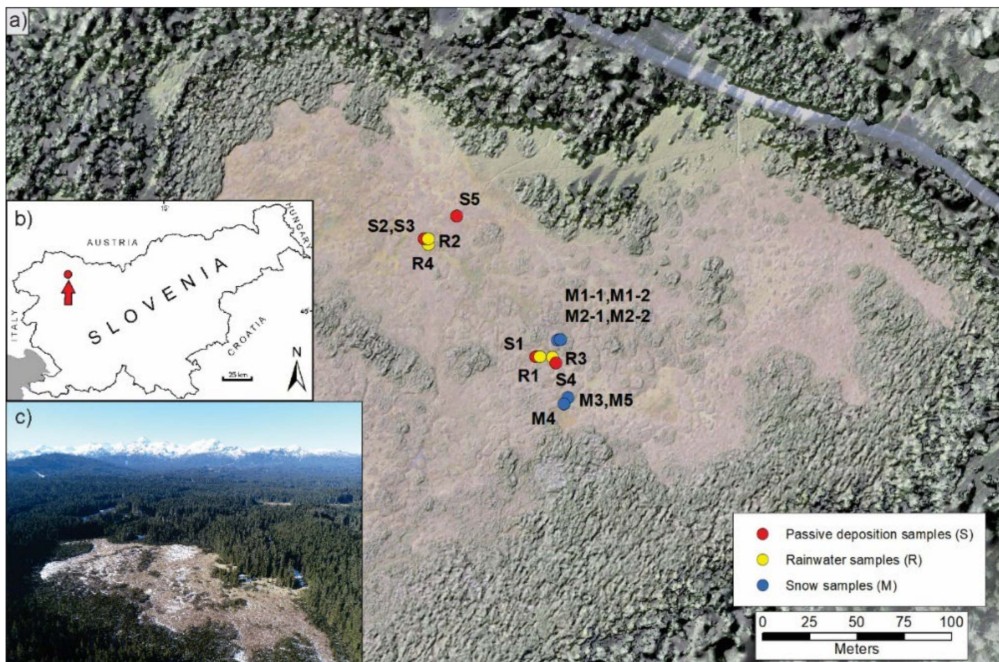

**Figure 1.** (**a**) Sampling locations in the middle and northwest of the Šijec bog with labelled samples. Orthophoto and topographic map was created with unmanned aerial vehicle (UAV). (**b**) Location of the Pokljuka plateau in Slovenia. (**c**) Aerial view of the Šijec bog and the Pokljuka plateau, showing the general topography of the plateau. Red dots marked with S represent Passive deposition samples locations, yellow dots marked with R represent Rainwater samples locations and blue dots marked with M represent snow samples locations.

Atmospheric deposition is an important process that removes gases and particles from the atmosphere. Dry deposition is the free sedimentation of atmospheric particulate matter (PM) and trace gases directly from the atmosphere. Wet deposition is the process whereby atmospheric gases and particulate matter mix with suspended water in the atmosphere and are then washed out as precipitation. Dry deposition and particulate matter in rain and snow were investigated in presented research. The main objective of this

study is to chemically and morphologically describe atmospheric particles deposited on the Šijec peat bog, as well as to determine the main possible sources and influence of other factors contributing to the mineral matter, such as dust events. This will provide a better understanding of the recent mineral matter input to this peatland that will later be related to a study of mineral matter within peat and water of the Šijec bog and its surroundings. Additionally, particle characterization helps to evaluate anthropogenic influence on the Pokljuka plateau.

## 2. Materials and Methods

### 2.1. Dust Events at Study Site in 2020 and 2021

Several dust events were observed in years 2020 and 2021. We sampled atmospheric deposition after the March 2020 and February 2021 dust events (Figures 2–4). The 27th–28th of March 2020 dust event resulted in elevated PM10 (particulate matter with diameter of less than 10 μm) levels throughout Slovenia, with measured values exceeding 100 μg/m$^3$ (Figures 2a and 4a) [28]. PM2.5 (particulate matter with diameter of less than 2.5 μm) levels were elevated, though not as much as the PM10 (Figure 2b). PM10 and PM2.5 levels gradually decreased in the following days. The highest PM10 and PM2.5 levels were detected in central and western Slovenia, respectively. Airborne particles were transported to Europe from Central Asia as part of an anomalous dust event [29].

After the 15th of February 2021, levels of PM10 over Slovenia often exceeded the daily limit (50 μg/m$^3$) [30]. Emissions from increased heating and traffic sources became more pronounced due to meteorological inversion and the absence of strong winds and precipitation that prevents the dilution of emissions. Particulate matter concentrations from 19th to 21st of February were elevated due to the inflow of polluted air from northern Italy (measured at 4 stations in western Slovenia: Koper, Nova Gorica, Solkan and Otlica) [30]. Additionally, a dust event originating from North Africa [16] occurred between 23rd to 26th of February 2021. Prior to this event was a larger dust event in parts of Europe in early February 2021 [16], although it was not detected in Slovenia [30]. The February 2021 event was not as pronounced as the 2020 event and has resulted in slightly increased values of PM10 (Figures 3a and 4b), while PM2.5 values were less affected as observed on Figure 3b.

### 2.2. Sampling and Sample Preparation

On the Šijec bog, we collected dry and wet atmospheric deposition following previously established techniques for collecting passive deposition [31], snow [31–33], and rainwater, which we slightly adapted to use in the relatively remote area of the Pokljuka plateau.

Five **passive deposition samples** were collected directly on the adhesive carbon tape with an area of 25 mm$^2$. SEM/EDS aluminum stubs with carbon tape were installed at two locations on the peatland at an approximate height of 180 cm (Figure 1) in October 2020 (2 samples in NW and 1 sample in the middle of peatland, collected after 12 days of exposure; Table 1) and February 2021 (2 samples collected after 8 days of exposure; Table 1). During the period when carbon tape was exposed in October 2020, the average temperature was 2.4 °C with 100 mm of precipitation. During February 2021 sampling, there was no precipitation, and the average temperature was −0.3 °C. In both 2020 and 2021, the predominant wind direction was from north and partly south–southeast, with both directions having strong winds. In February, there was an additional increase in slow winds (less than 1 m/s) from the west direction. SEM/EDS stubs with collected dry atmospheric deposition were carbon-coated before the analysis to obtain conductivity of the samples.

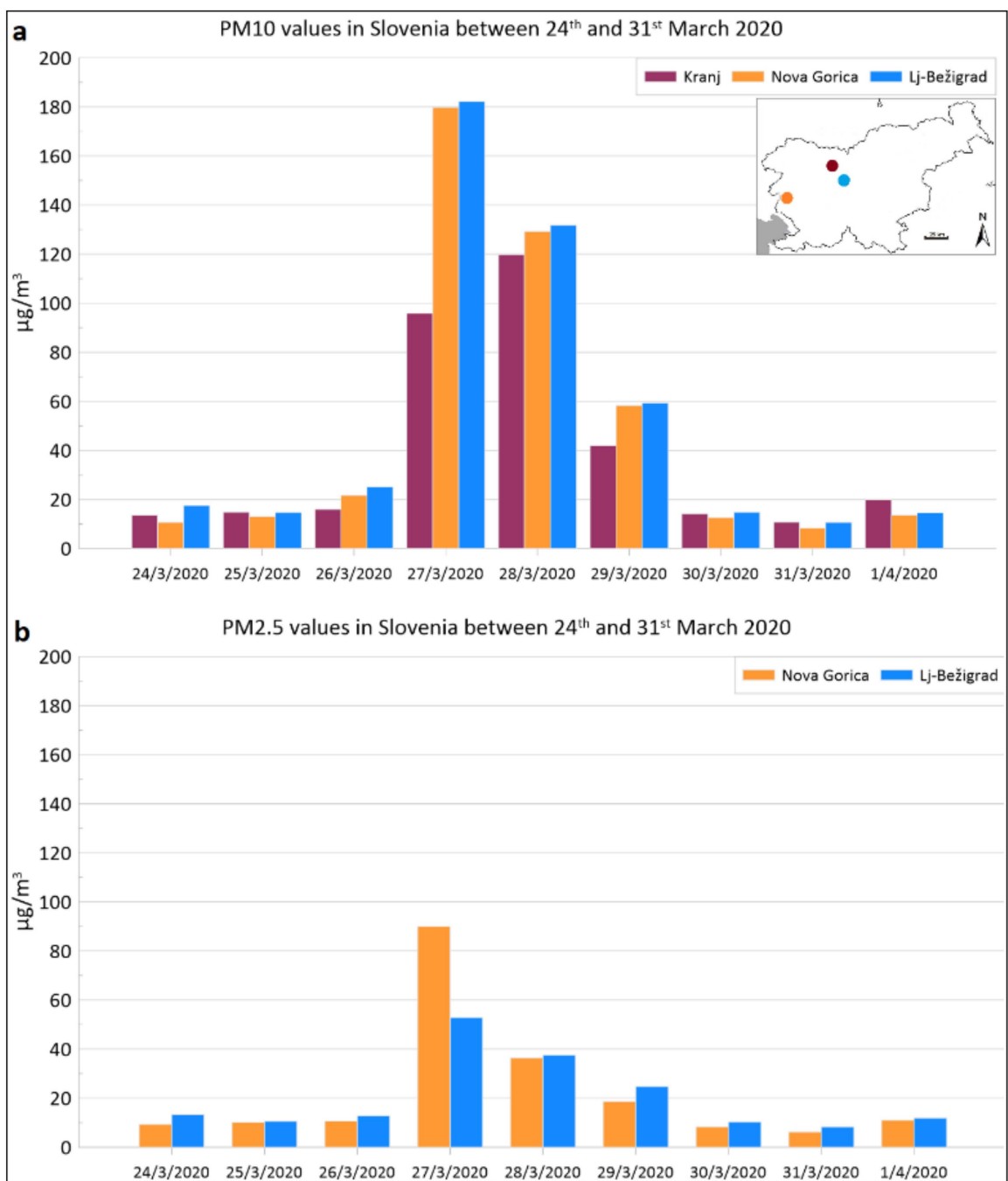

**Figure 2.** Daily PM10 (**a**) and PM2.5 values (**b**) in Slovenia measured in 3 stations (Ljubljana, Nova Gorica and Kranj) between 24th and 31st of March 2020, showing the influence of the 2020 dust event. Data was provided by Slovenian Environment Agency (ARSO).

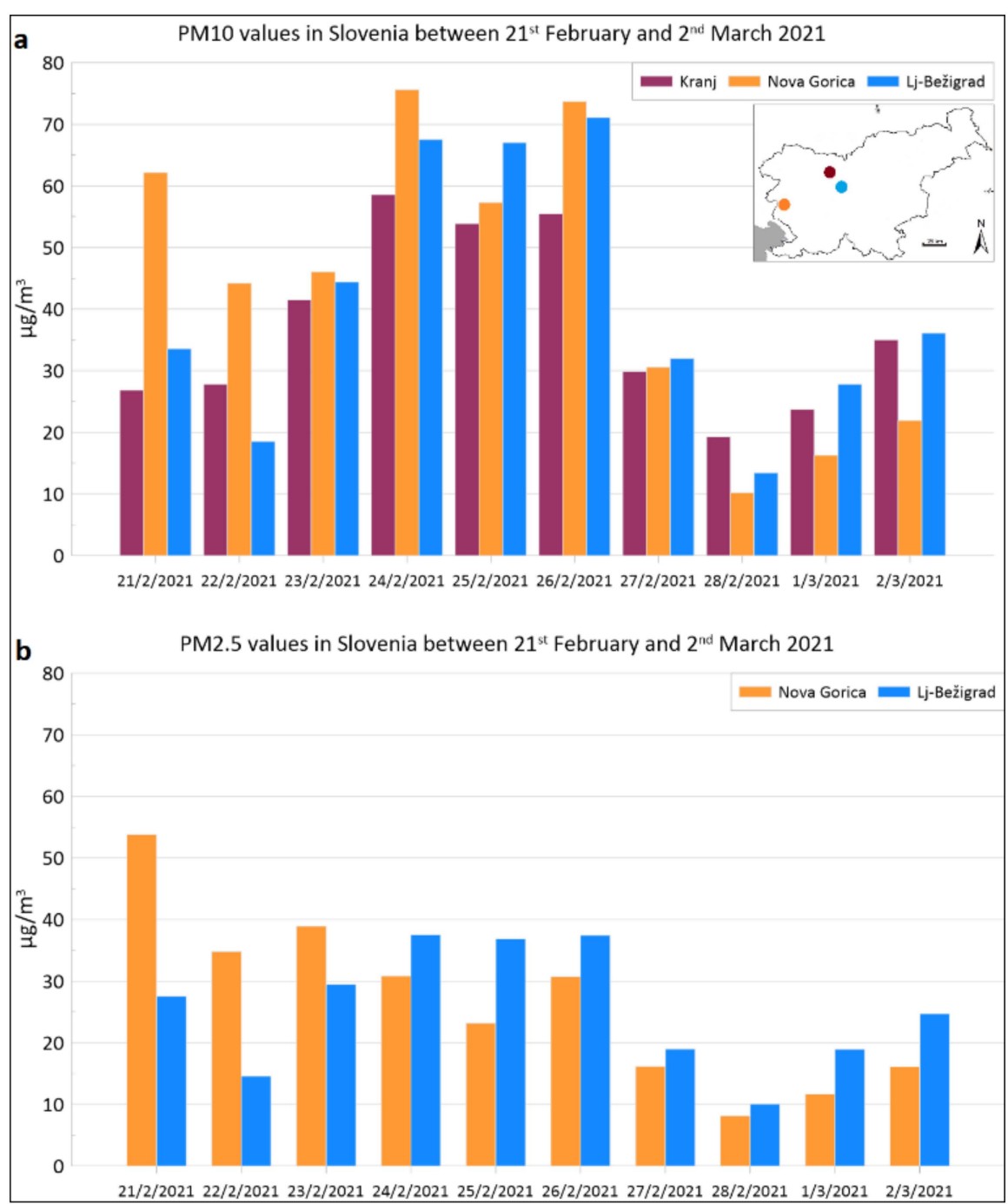

**Figure 3.** Daily PM10 (**a**) and PM2.5 values (**b**) in Slovenia measured in 3 stations (Ljubljana, Nova Gorica and Kranj) between 21st of February and 2nd of March 2021, showing the influence of the 2021 dust event. Data was provided by Slovenian Environment Agency (ARSO).

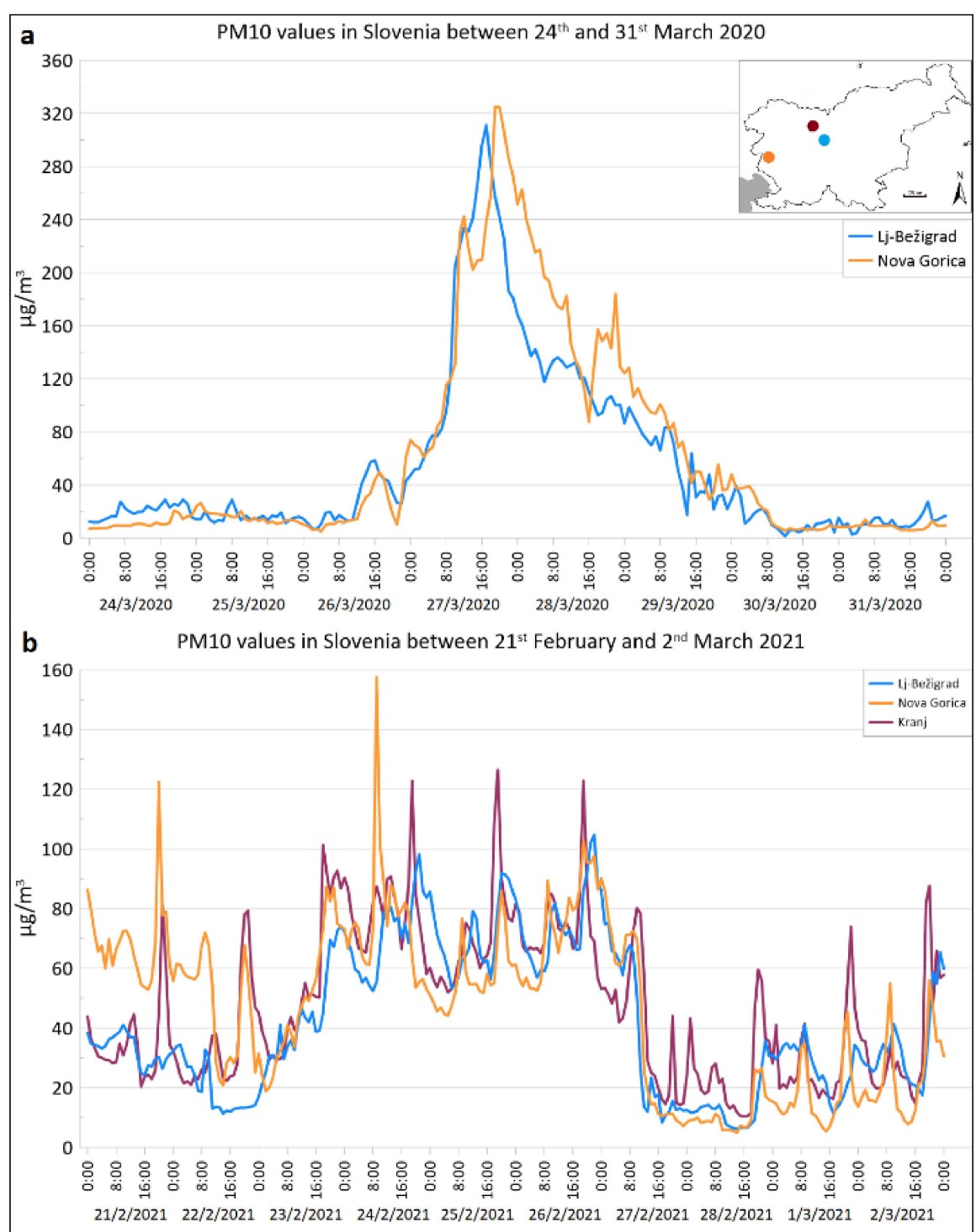

**Figure 4.** Hourly PM10 values in Slovenia measured in 3 stations (Ljubljana, Nova Gorica and Kranj) during dust event periods in 2020 (**a**) and 2021 (**b**). Data was provided by Slovenian Environment Agency (ARSO).

**Table 1.** List of collected samples, dates of sampling, and sampling locations.

| Sample | Code | Date of Sampling | Location (See Figure 1) | Depth in Case of Snow Profile Sampling |
|---|---|---|---|---|
| Passive deposition 2020 | S1 | 8.10.–20.10.2020 | Middle | - |
| | S2 | 8.10.–20.10.2020 | NW part | - |
| | S3 | 8.10.–20.10.2020 | NW part | - |
| Passive deposition 2021 | S4 | 24.2.–4.3.2021 | Middle | - |
| | S5 | 24.2.–4.3.2021 | NW part | - |
| Rainwater 2020 | R1 | 8.10.–25.11.2020 | Middle | - |
| | R2 | 8.10.–25.11.2020 | NW part | - |
| Rainwater 2021 | R3 | 3.8.–23.9.2021 | Middle | - |
| | R4 | 3.8.–23.9.2021 | NW part | - |
| Snow 2020 | M1-1 | 31.3.2020 | Middle, upper layer | 0–20 cm |
| | M1-2 | 31.3.2020 | Middle, lower layer | 20–32 cm |
| | M2-1 | 31.3.2020 | Middle, upper layer | 0–20 cm |
| | M2-2 | 31.3.2020 | Middle, lower layer | 20–32 cm |
| Snow 2021 | M3 | 24.2.2021 | Middle | 0–5 cm |
| | M4 | 24.2.2021 | Middle, snow dune | 0–5 cm |
| | M5 | 4.3.2021 | Middle | 0–5 cm |

**Wet atmospheric deposition** was collected in the form of 4 rainwater and 7 snow samples (Table 1). The **rainwater** samplers (20-litre collector with a built-in funnel with a radius of 160 mm and a surface area of $0.02 \ m^2$) were installed at two locations in October 2020 and August 2021 (Figure 1) and collected after approximately two months of exposure (Table 1). During rainwater sampling in October 2020, 4.3 L were collected at the first location (R1) and 3.7 L at the second location (R2). Similarly, we collected 3.6 L (R3) and 3.8 L (R4) of rainwater in August 2021 (Table 2). Collected rainwater samples were filtrated using isopore membrane polycarbonate filters (<0.6 μm) (Merck KGaA, Darmstadt, Germany) to obtain particulate matter. Weather conditions were obtained from the Rudno polje meteorological station, located 5 km from the Šijec bog at similar elevation. The average temperature during the 2020 and 2021 sampling periods was 2.6 °C and 11.8 °C, respectively. Total precipitation was 185 mm for both sampling periods, which is consistent with the amount of collected rainwater. In October 2020, part of the precipitation during sampling period was deposited as snow, which affected the sample yield, which was larger compared to the data measured at the meteorological station. The measured wind directions and speeds were similar during both periods, with predominantly stronger winds from the north and south, and weak winds from the west direction. No major dust event was recorded during either period.

**Table 2.** Physico-chemical parameters (pH, electrical conductivity) measured in snowmelt and rainwater samples and volume of water samples.

| Water Sample | Code | Volume of Water Sample (L) | pH | EC (μS/cm) | T (°C) |
|---|---|---|---|---|---|
| Rainwater 2020 | R1 | 4.3 | - | - | - |
| | R2 | 3.7 | - | - | - |
| Rainwater 2021 | R3 | 3.6 | 7.14 | 23.97 | 17.0 |
| | R4 | 3.8 | 6.27 | 7.54 | 16.6 |
| Snowmelt 2020 | M1-1 | 1.85 | 6.32 | 5.99 | 17.0 |
| | M1-2 | 5.5 | 6.30 | 7.57 | 16.9 |
| | M2-1 | 1.65 | 6.34 | 5.51 | 17.0 |
| | M2-2 | 5.5 | 6.41 | 6.69 | 16.9 |
| Snowmelt 2021 | M3 | 3.6 | 5.52 | 2.96 | 16.0 |
| | M4 | 3.2 | 5.69 | 3.71 | 15.6 |
| | M5 | 3.8 | 5.78 | 3.27 | 15.8 |

In the 2020 and 2021 winters, we collected deposited **snow** in an area of 0.5 m × 0.5 m in the middle of the peatland, avoiding sampling in the vicinity of trees and bushes (Figure 1). The depth of sampling was determined based on observations in the snow profile and weather conditions.

In March 2020, we collected 4 snow samples from the middle part of the peatland (Figure 1). Two samples (M1-1 and M2-1) were collected from the upper, fresh snow layer (0–20 cm) that was deposited 2 days prior to the sampling. The other two samples (M1-2 and M2-2) were collected from the lower, denser layer (20–32 cm) that was deposited 30 days before sampling, with few shorter periods of snowfall in between. Due to the more compacted lower layer, a smaller surface area of 0.5 m × 0.25 m was used for its sampling. There was a major dust event prior to snow sampling that caused an increased concentration of airborne and deposited particles [28]. The particulate matter from the dust event was deposited on top of the denser snow layer and was later covered with fresh snow. During the 30-day period, total precipitation was 180 mm, 25 mm of which formed the upper layer of the sampled snow. The average temperature was −1.2 °C for a 30-day period and −3.8 °C for the two-day period prior to sampling. The predominant wind direction during the 30-day period was from the north, with few events of south winds. Wind speed was greater from the north direction, with an average speed of 2.3 m/s and maximum wind gusts of 17.1 m/s. In the two-day period, there were only stronger winds from north direction (average speed of 2.3 m/s).

In February 2021, two samples of the upper 5 cm of snow (M3 and M4) were collected 18 days after the last snowfall. Due to high winds prior to the sampling, solid particles had accumulated on the sides of snow dunes, which we observed in the field. We sampled both the snow with visibly accumulated solid particles and the snow without them. In March 2021, an additional snow sample was collected (depth 0–5 cm) after an event of increased atmospheric dust deposition [30]. There was no new snowfall between the two sampling periods. The February 2021 dust event reached Slovenia at the time of the first sampling and persisted for 3 days (Figure 3). The average temperature was −3 °C and the total precipitation was 135 mm. During the 18 days prior to the first sampling, the predominant wind direction with the strongest winds was north (average speed 2.1 m/s with wind gusts of 12.3 m/s) and NNW (average speed 1.6 m/s with wind gusts of 16.7 m/s). There were also multiple events of relatively strong winds from the south and weak winds from the west direction. Before the second sampling, there was an increase in the wind from the west direction.

After sampling, snow was melted at room temperature and yielded 3–6 L of snowmelt water in the denser samples (lower layer March 2020 and all 2021 samples) and 1.5–2 L in the surface samples (upper layer March 2020) (Table 2). As well as rainwater, snowmelt water was filtrated through 0.6 μm polycarbonate filters to separate particulate matter and water sample. After filtering, electrical conductivity (EC) and pH were measured in the water samples using Thermo Scientific Orion Star A329.

The particulate matter obtained from rainwater and snowmelt on the filters was dried at room temperature and mounted on a double-sided carbon tape with a surface area of 25 mm$^2$ and coated with a thin layer of carbon for conductivity. As the samples were placed on carbon tape and carbon-coated, the presence of C in individual particles could not be determined. Particulate matter gained from all samples (7 snow, 4 rainwater, and 5 passive deposition samples), was analyzed with scanning electron microscopy coupled with energy dispersive spectroscopy (SEM/EDS) using JEOL JSM 6490LV SEM (JEOL Ltd., Tokyo, Japan) coupled with an Oxford INCA PentaFETx3 Si(Li) detector (Oxford Instruments Analytical, Ltd., High Wycombe, UK) and INCA Energy 350 processing software (Oxford Instruments Analytical, Ltd., High Wycombe, UK) installed at Geological survey of Slovenia.

### 2.3. SEM/EDS Analysis

The SEM/EDS analysis was performed in high vacuum at an accelerating voltage of 20 kV, spot size 48–50, and a working distance of 10 mm. Particles in passive deposition

samples are sparse, so all particulate material gathered on carbon tape was considered for analysis. In contrast, for the analysis of particles from snowmelt water and rainwater, several areas (fields of view) on the carbon tape were randomly selected where individual particles were analyzed. The amount of particulate matter in precipitation samples was high, with particles were often overlying each other.

Surface EDS analysis (mapping) was performed for all particulate matter samples obtained from precipitation to obtain the general ratios of specific mineral groups by visually comparing selected fields of view with Charts for Estimating Percentage Composition of Rocks and Sediments [34]. Magnification for EDS elemental mapping was adjusted for each sample but kept similar for comparison between different samples. We used magnifications ×300, ×600, and ×800 (in S1.1.-S1.3.). In each selected area, we analyzed individual particles by determining their size, shape, morphology, and chemical composition. Particle size was measured along its longest dimension. Shape and morphology were determined by the degree of rounding [34].

## 3. Results

### 3.1. Electrical Conductivity and pH of Snowmelt and Rainwater

Snowmelt water has a very low electrical conductivity and slightly acidic to neutral pH (Table 2). We determined the lowest values of both parameters in 2021 snowmelt samples (M3, M4 and M5), with conductivity values around 3.0 μS/cm and pH values between 5.5 and 5.8. In 2020 snowmelt samples, pH is approximately 6.3 for all samples, but conductivity values vary between the upper and lower layers of snow. Lower values of electrical conductivity (5.5 and 6.0 μS/cm) are found in samples from the upper snow layer (M1-1, M2-1), while higher values (6.7 and 7.6 μS/cm) are observed in the compacted lower snow layer (M1-2 and M2-2).

Rainwater sample R4 has a similar pH (6.3) and electrical conductivity (7.5 μS/cm) as snowmelt water obtained from the lower snow layer (M1-2 and M2-2), while the rainwater sample R3 has much higher electrical conductivity (24 μS/cm) and a pH value of 7.1. Long exposure (few months) on the peatland allows more particles to dissolve in the water, affecting pH and electrical conductivity [35]. Dissolution of carbonates originating from the Šijec bog surroundings presents the most probable cause for the increase in pH and electrical conductivity in the rainwater samples.

### 3.2. SEM/EDS Characterization of Particles

3.2.1. Passive Deposition Samples

Particles deposited on the adhesive carbon tape of the passive samplers are generally sparse, although the number of particles varies between samples. Samples collected in 2021 following a dust event in February [30] contain more particles than samples collected during average conditions in 2020. However, the difference in particle quantity may also be due to different seasonal conditions during sampling. The number of particles in passive deposition samples is smaller in comparison with other sample mediums. However, passive deposition is a good indicator of what particle types are present in a certain time period.

Based on mineral composition, assessed on the basis of determined chemical composition, we classify analyzed particles into the following main groups: quartz, aluminosilicates, carbonates, organic matter, and Fe-oxyhydroxides. Aluminosilicates and quartz ($SiO_2$) predominate in all passive deposition samples. Among the aluminosilicates we further distinguish two groups: (1) clay minerals consisting of mainly Si, Al, and O, with smaller contents of other elements such as Fe, Mg, and K, and (2) aluminosilicates consisting of Si, Al, and a higher content of K, Na, and Ca, as commonly found in feldspar minerals, as well as Mg and Fe, which are common in the olivine mineral group. Since it is not possible to determine the degree of weathering based on composition alone, the latter group also includes weathering products of feldspar and olivine mineral groups.

Both silicate groups, quartz and aluminosilicates, have a predominantly angular shape and a wide range of sizes from 0.5 μm to 50 μm (Figure 5a,b). Particles in the 2021 dust

event are smaller than particles in the 2020 samples. The average size of quartz particles in 2020 samples is 11.7 μm, while the average size of quartz particles from 2021 is 5.8 μm. A similar difference in size is observed for aluminosilicate (clay mineral) particles. Their average size from 2020 is 20.4 μm, while clay mineral particles from 2021 have an average size of 6.7 μm. The size comparison of other aluminosilicate particles (feldspar, olivine, and other minerals) shows a small difference, with an average size of 13.8 μm in 2020 and 10.2 μm in 2021.

Carbonates are present in all passive deposition samples (Figure 5c,d). Particles consisting of Ca-Mg-C-O (dolomite) are more abundant than Ca-C-O particles (calcite, aragonite). The average size of carbonate particles in 2020 and 2021 was 7 μm and 12.3 μm, respectively.

Since the Šijec bog is located in a densely forested area of the Pokljuka plateau (Figure 1), particles of organic matter are often found in the samples. Although not abundant, they are usually represented as larger, (5–10 μm) rounded, spherical (Figure 5e,f), or elongated particles, which may also contain other elements (Si, Al, Ca) in small amounts.

Iron-oxyhydroxides, some of which contain Si, Al, Mn, Mg, and K in small amounts, represent a smaller but significant part of the particles. Their average size is small, 2.6 μm and 4.6 μm in the 2020 and 2021 samples, respectively. They are predominantly angular, with spheres rarely present. Spheres, irregular shapes, agglomerates, and shavings, which are also present in our samples, are typical indicators of anthropogenic sources according to Gaberšek and Gosar [31,36].

Other particles found in the passive deposition samples are mostly metal-bearing particles, such as Ti-oxides (e.g, rutile), Pb-oxides/carbonates and Pb-sulphides (e.g., galena), Fe-sulphides (e.g., pyrite), particles containing multiple metals (commonly including Fe, Mn, Ti, Ni, and Cr, Figure 5g) as well as other particles, such as salt (NaCl, Figure 5h). Their shape is either spherical or angular and are mostly small (<5 μm).

### 3.2.2. Particles from Rainwater

EDS elemental mapping was first performed on each rainwater sample to determine the general chemical composition of the particles which was then used to assess their mineral composition (examples in S1.1 and S1.3). The mineral composition of particles in the rainwater samples (R1–R4) and their estimated percentage content are presented in Table 3. Most of the particles (>50%) are silicates, which we divide into quartz and aluminosilicate particles. We further subdivided the latter into clay minerals and other aluminosilicate minerals, the most important being feldspar and olivine group and their weathering products. Silicate particles, both quartz and aluminosilicates (Figure 6a–d) are predominantly angular and have a wide range of sizes, 0.5–20 μm and 0.5–50 μm in 2020 and 2021 samples, respectively. The samples collected in 2021 contain a higher number of larger particles (>10 μm) in contrast to the 2020 samples, where smaller particles predominate.

**Table 3.** Estimated percentage content of main particle groups in rainwater and snowmelt water samples. The content of other particles is less than 5% but varies between samples and was not included in this calculation.

| Sample Code | Quartz (%) | Aluminosilicates (%) | Carbonates (%) | Organic Matter (%) | Fe-Oxyhydroides (%) |
|---|---|---|---|---|---|
| R1 | 20 | 40 | 15 | 15 | 10 |
| R2 | 25 | 40 | 15 | 15 | 5 |
| R3 | 30 | 35 | 15 | 10 | 10 |
| R4 | 15 | 45 | 20 | 10 | 10 |
| M1-1 | 25 | 30 | 20 | 15 | 10 |
| M1-2 | 40 | 30 | 20 | 5 * | 5 |
| M2-1 | 25 | 40 | 15 | 10 | 10 |
| M2-2 | 35 | 45 | 5 | 5 * | 10 |
| M3 | 40 | 35 | 5 | 10 | 10 |

**Table 3.** *Cont.*

| Sample Code | Quartz (%) | Aluminosilicates (%) | Carbonates (%) | Organic Matter (%) | Fe-Oxyhydroides (%) |
|---|---|---|---|---|---|
| M4 | 35 | 30 | 10 | 15 | 10 |
| M5 | 40 | 40 | 5 | 10 | 5 |

\* Organic particles are covered by small silicate particles. As percentage is determined based on coverage percentage of the field during EDS elemental mapping, they were not detected.

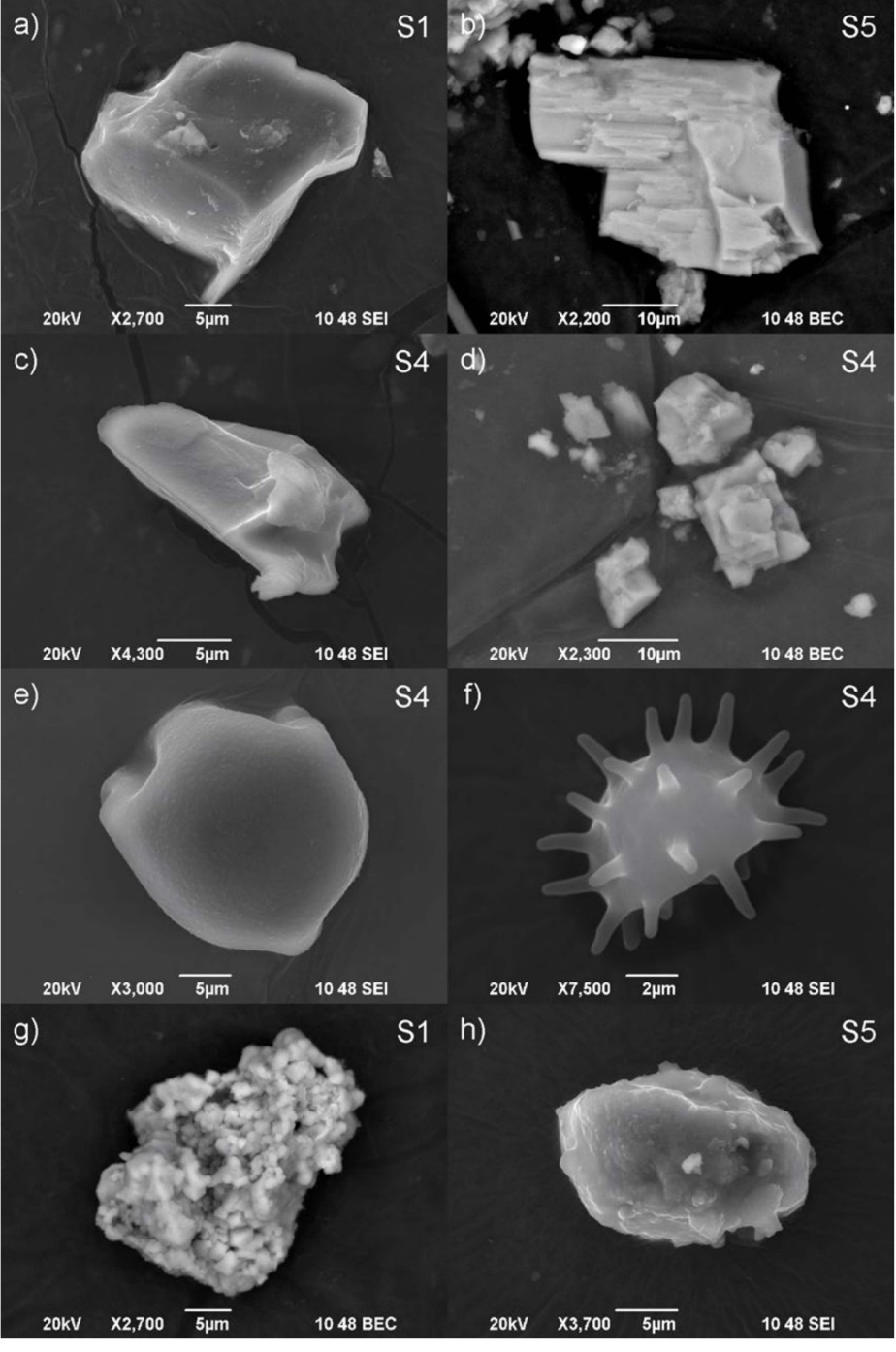

**Figure 5.** Particles from passive deposition samples. Presented images are taken in SEI or BSE modes as indicated in the lower right corner of images. The sample code is marked in the upper right corner of each figure. (**a**) angular Si-O (quartz); (**b**) angular Si-Al-Ca-Mg-O particle (aluminosilicate); (**c**) and (**d**) Ca-Mg-C-O (dolomite); (**e**) and (**f**) organic particles, spores; (**g**) irregularly shaped Fe-Cr-O agglomerate; (**h**) Na-Cl (salt). Some spectra are provided in Supplementary Figure S2.

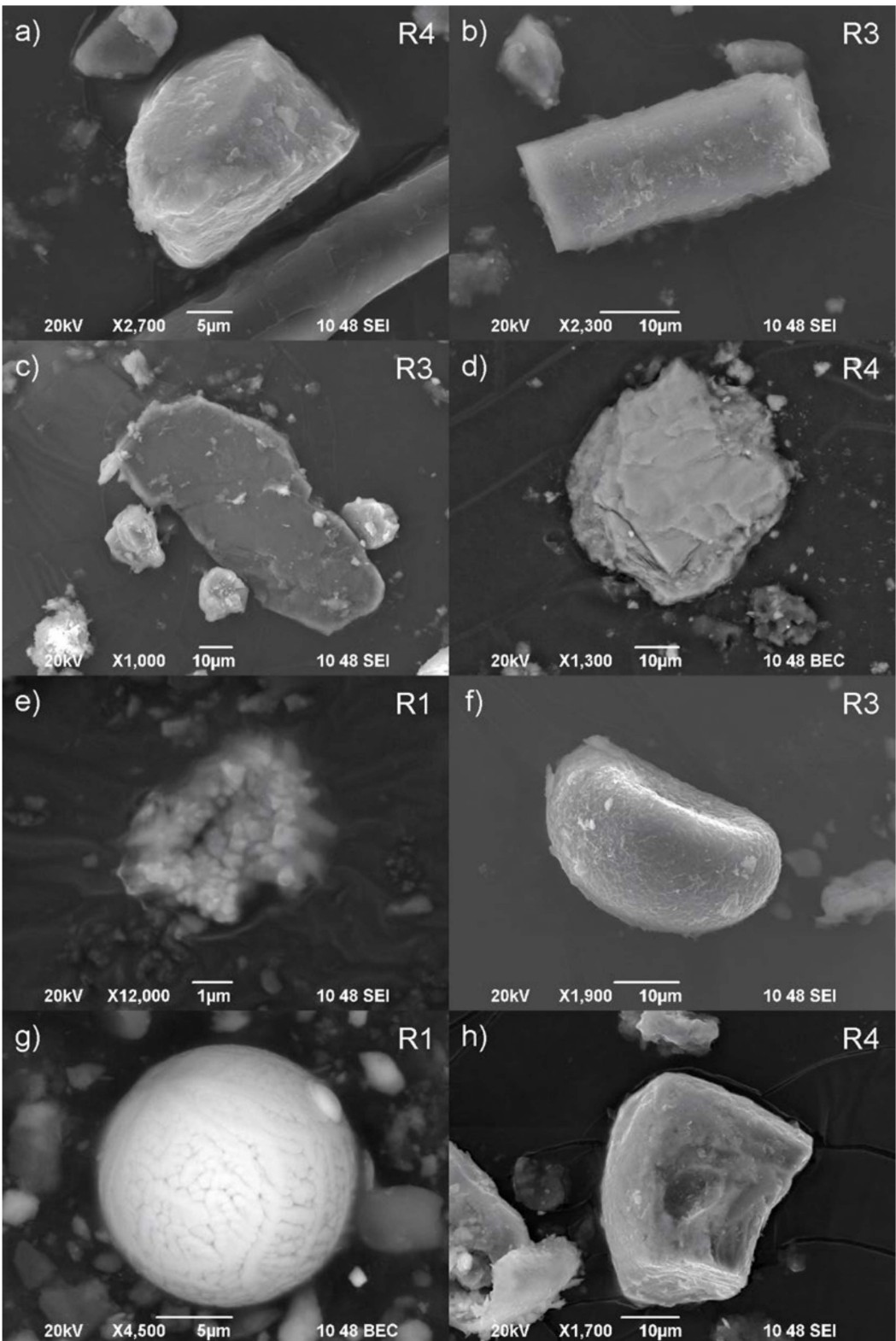

**Figure 6.** Particles from rainwater. Presented images are taken in SEI or BSE modes as indicated in the lower right corner of images. The sample code is marked in the upper right corner of each figure. (**a**,**b**) angular Si-O (quartz); (**c**) aluminosilicates with simple composition (Al-Si-O); (**d**) angular aluminosilicate (Si-Al-K-O); (**e**) crystalline structure of Ca-C-O particle (calcite or aragonite); (**f**) organic particle, spore; (**g**) Fe-O (Fe-oxyhydroxide) skeletal-dendritic sphere; (**h**) angular Zr-Si-O particle (zircon).

Carbonates, Ca-C-O and Ca-Mg-C-O (representing minerals calcite, aragonite, and dolomite), represent 15%–20% of the particles. The content of carbonate particles, particularly calcite particles, may be influenced by their dissolution in the atmosphere or after deposition in the sampler. Furthermore, Ca-C-O particles often appear as small, (<5 μm) irregularly shaped crystals (Figure 6e). The crystalline structure of calcite/aragonite particles most likely shows that they crystallized from the residual rainwater during drying of the filter papers. In comparison, dolomite particles are often present as larger, angular particles.

Organic matter represents approximately 10%–15% of the particles from the rainwater (Table 3). The particles are mostly rounded or spherical and generally larger than 5 μm (Figure 6f). Organic particles in the 2020 samples are smaller (average 6.4 μm) compared to the 2021 particles (average 17.4 μm). Organic matter also occurs in the form of filaments and irregularly shaped particles, or it may partially coat mineral particles.

Iron-oxyhydroxides represent 10% of the particles in all samples. The samples from rainwater contain a large number of Fe-oxyhydroxide spheres (Figure 6g), as well as a significant portion of angular Fe-oxyhydroxides. Their size varies and reaches up to 40 μm. However, the average size of Fe-oxyhydroxides is 8.0 and 8.8 μm in 2020 and 2021 samples, respectively.

Other particles include Zr-Si-oxides (e.g., zircon, Figure 6h), Ti-oxides (e.g., rutile), Zn-oxides/carbonates, Pb-oxides/carbonates and sulphides (e.g., galena), Fe-sulphides (e.g., pyrite), Ba-sulphates (e.g., baryte), rare earth minerals (most common are Ce-(La-Nd)-P-O particles), and other Fe, Mn, Ni, Cr, and Zn bearing particles. They occur as spheres or in angular and irregular shapes. They have various sizes with the largest reaching up to 60 μm, though most of them are smaller (<5 μm).

### 3.2.3. Particles Deposited in Snow

Similar to previous sample types, silicate particles predominate in all snow samples, both from 2020 and 2021. In the 2020 samples, there is a clear difference between the lower and upper layers in terms of silicate content (Table 3). Compared to the upper layer with around 60% of silicates, the lower layer has a higher silicate content (>70%). This is mainly due to a higher quartz content (40% and 35%), whereas aluminosilicate content remains similar as in the upper layer. The upper layer contains approximately 25% quartz and 30–40% aluminosilicates. In the 2021 samples, the silicate content is similar to that of the lower snow layer in 2020, with quartz content ranging from 35%–40% and the aluminosilicate content ranging from 30%–40% (Table 3).

In the upper layer of snow from 2020, quartz particles are angular, and ranging from 0.5 μm to 20 μm. Smaller aluminosilicate particles (<5 μm) predominate, though the largest particles can reach up to 100 μm. They are mostly angular. In samples from the lower snow layer, the size range of angular quartz particles is 0.5–50 μm. Compared to the particles in the upper layer (average size of 14.1 μm), the aluminosilicate particles in the lower snow layer have a smaller range with an average value of 11.0 μm. The average value of the latter is biased as not all small (0.5 μm) particles were measured and included in the calculation. The angular quartz particles in samples without (samples M3 and M4) and with dust event recorded (sample M5) from 2021 have similar size ranges (Figure 7a). Their average values are 12.6 μm and 11.8 μm, respectively. A difference is apparent for the aluminosilicates, where the average particle size is 18.6 μm before the dust event and 9.5 μm during the event. Their shape is predominantly angular, with few rounded or spherical particles (Figure 7b–e).

The samples from 2021 show a higher number of large particles (>10 μm) than in the 2020 samples. A similar observation was noted by the Slovenian Environment Agency for the period of the dust event [30]. The difference between the amount of smaller and larger particles becomes more evident when comparing the data from 2020 and 2021. Although small particles were abundant in both years, their content was greater in 2020, especially in the lower snow layer with records of the March 2020 dust event.

The carbonate content in the 2020 samples is higher (15%–20%) than in the 2021 samples (5%–10%), except for one sample (M2-2) from the lower snow layer (5%). This can be due to dissolution of carbonate particles in snowmelt water, similar as in rainwater, affecting the carbonate content in the sample. Carbonate particles, Ca-C-O, and Ca-Mg-C-O have similar size and shape in all snow samples. Their average size is 8 μm and they occur in angular form or Ca-C-O (calcite or aragonite) crystals that crystallized on the filter paper during drying (Figure 7f,g). Dolomite particles are generally larger, reaching a size of >10 μm, while calcite represents smaller particles.

The content of the organic particles is similar in all samples, ranging from 10% to 15%. The exceptions are the lower snow layer samples from 2020, where the content is estimated at 5%. This is due to the large amount of smaller silicates covering the organic particles, and the estimation of the percentage composition [34], relies on visible particles. The organic matter is present as large (average 21.3 μm), rounded particles (Figure 7g,h), as filaments, or as coatings of organic matter over mineral particles.

Iron-oxyhydroxides are present in all snow samples and account for 5%–10% of the particles. The angular shape predominates, but there are also Fe-oxyhydroxides in spherical shape (Figure 8a,b), irregular shape, and agglomerates. The sizes differ between the samples. In the 2020 samples, the upper snow layer contains particles with an average size of 5.3 μm, while the lower snow layer contains particles with an average size of 8.7 μm. The average size of particles in the lower layer may be biased due to large amount of small particles (<1 μm), causing them to overlay each other. This also affects EDS detection as it is unable to distinguish individual particles. Therefore, identification of the chemical composition of smaller particles can be influenced by a high content of Si and Al from nearby aluminosilicates.

In samples collected before the dust event in 2021, the average size of Fe-oxyhydroxides is 11.8 μm, while the average size of the same particle type from the 2021 dust event samples is 6.4 μm.

Other particles in the snow (Figure 8) include mainly Fe-oxyhydroxides containing other elements, such as Ca, Ti, Cr, Fe-sulphides (e.g., pyrite), Zr-Si-oxides (e.g., zircon), Ti-oxides (e.g., rutile), Ba-sulphates (e.g., baryte), rare earth minerals (Ce-La-Nd-P-O), and Pb-oxides/carbonates or Pb-sulphides (e.g., galena).

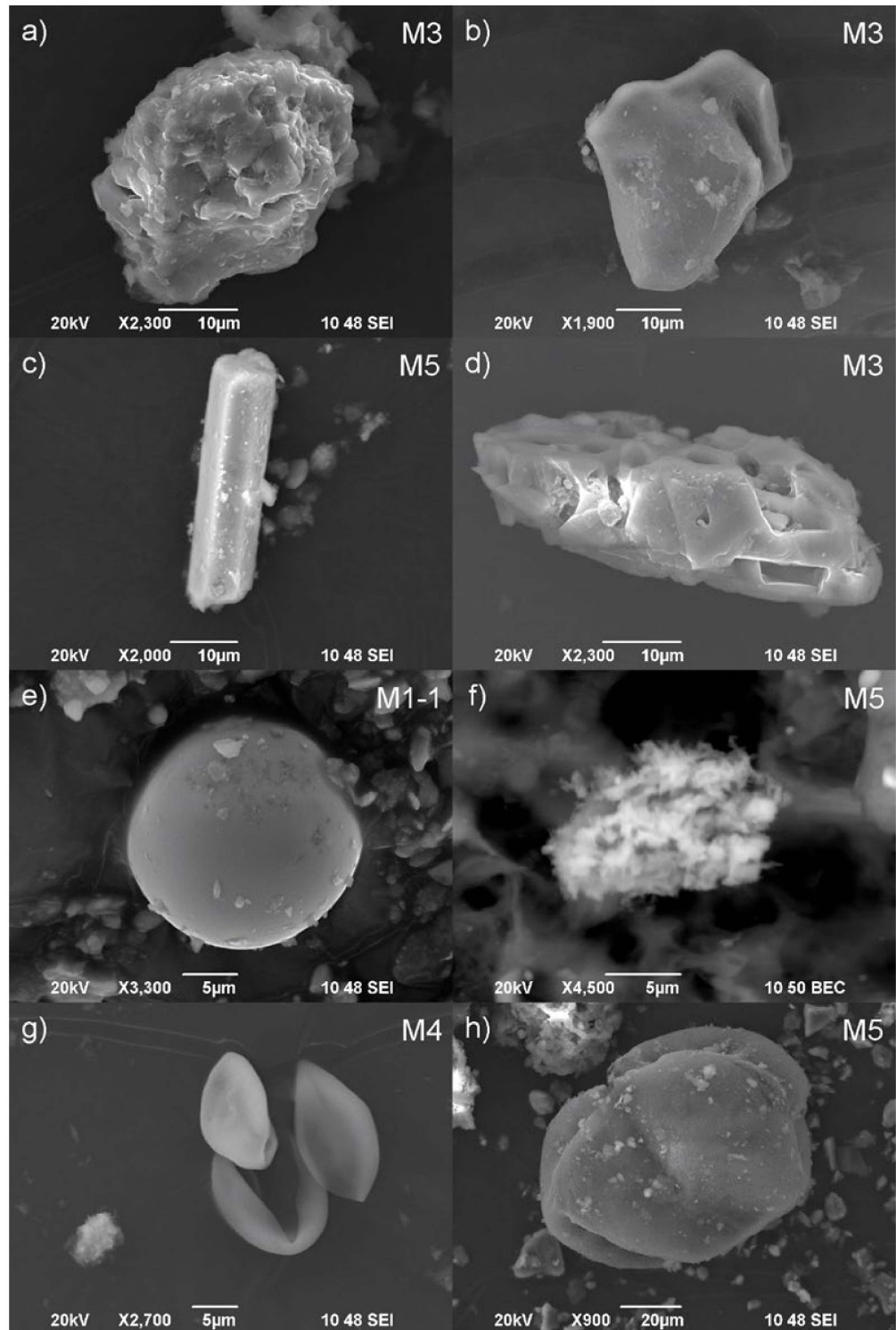

**Figure 7.** Particles from snowmelt water. Presented images are taken in SEI or BSE modes as indicated in the lower right corner of images. The sample code is marked in the upper right corner of each figure. (**a**) Si-O (quartz); (**b**) and (**c**) aluminosilicates with simple composition (Al-Si-O); (**d**) aluminosilicate (Si-Al-Na-O); (**e**) spherical aluminosilicate (Si-Al-K-Ca-O); (**f**) crystalline structure of carbonate (Ca-C-O, calcite or aragonite) with elevated content of Mg; (**g**) organic particles (spores on the right of the image) and crystalline structure of Ca-C-O (calcite or aragonite in the lower left corner of image); (**h**) organic particle spore covered with small aluminosilicate particles. Some spectra are provided in Supplementary Figure S3.

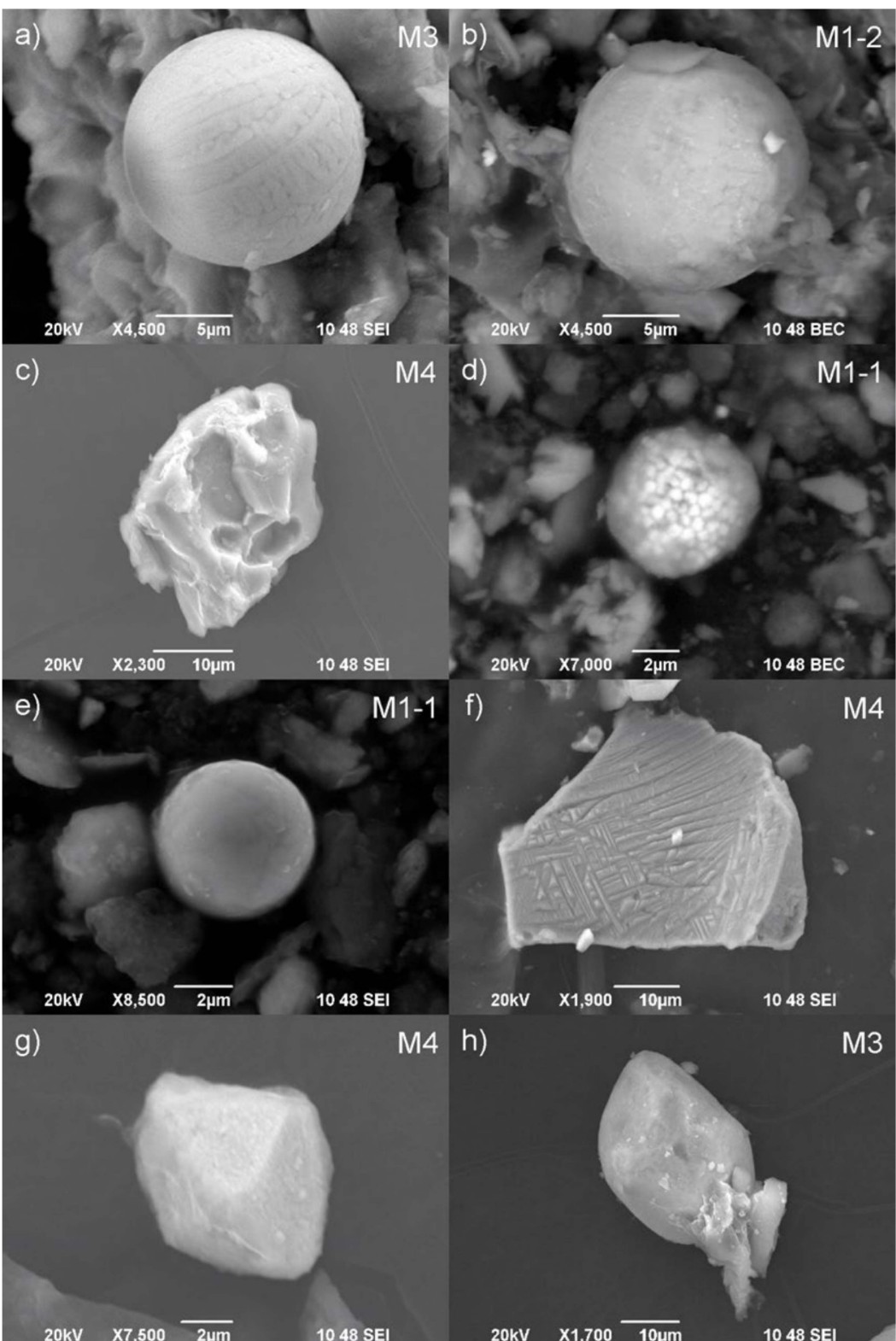

**Figure 8.** Metal particles from snowmelt water. Presented images are taken in SEI or BSE modes as indicated in the lower right corner of images. The sample code is marked in the upper right corner of each figure. (**a**) Skeletal-dendritic Fe-O sphere; (**b**) Fe-Cr-O sphere; (**c**) Fe-Ca-O particle; (**d**) Fe-S particle (framboidal pyrite); (**e**) Ti-O sphere; (**f**) Fe-Ti-O particle with visible ribs; (**g**) Pb-S (galena); (**h**) Zr-Si-O (zircon). Some spectra are provided in Supplementary Figure S3.

## 4. Discussion

### 4.1. Evaluation of Application of Various Atmospheric Deposition Mediums

Throughout our results it becomes evident that apart from weather conditions during sampling, obtained atmospheric deposition samples are influenced by the type of used mediums.

Passive deposition depends largely on weather conditions, as even weak winds can prevent smaller particles from free falling to the passive sampler. In the case of stronger winds, it is also possible that already deposited particles are removed from the carbon tape if they do not adhere to it. Particles deposited on the carbon tape are sparse, which decreases the usability of EDS elemental mapping for assessment of main mineral groups present in the samples and their proportions. General chemical composition of particles could therefore be obtained only through individual analysis of all deposited particles which is more time-consuming. Nevertheless, sampling of passive deposition proved to be a good indicator of increased atmospheric deposition during a dust event. Visual comparisons of samples from 2020 (normal conditions) and 2021 (dust event) show a larger amount of particles in the 2021 sample.

Preparation of passive deposition samples before the analysis with SEM/EDS requires only carbon coating, whereas particles from snowmelt and rainwater must first be separated from liquids (filtrating) and then transferred from filter paper to the carbon tape. Applying too many particles on the tape can cause partial or complete covering of underlying particles (e.g., samples M1-2 and M2-2 from lower snow layer in 2020). Additionally, while filtering rainwater and snowmelt, heavier particles settle on the filter faster than lighter particles, which remain suspended in the water for longer. This can cause uneven, gradual deposition on the filter. Therefore, only applying part of the particulate matter to the carbon tape can exclude certain particle groups or sizes.

Rainwater and snow samples represent both dry and wet atmospheric deposition. For rainwater samples, the rain (wet atmospheric deposition) is collected along with the dry atmospheric deposition in a single container exposed for the duration of a predetermined sampling period. In the case of snow sampling, we collect atmospheric deposition for the period between the last snowfall and sampling. As it is not always possible to distinguish different layers of snow, there is a possibility of sampling snow from previous snowfall. Therefore, in some cases the period of atmospheric deposition prior to sampling is not correctly determined. This is more important when obtaining samples from a specific period, such as a dust event.

After the snowfall, partial melting may occur, causing particles to move in the snow profile. Snow transport can occur by strong winds and is influenced by topography [37], where snow (and particles deposited within it) accumulate in snow dunes. This can create an uneven distribution of deposited snow. Micro-location of snow sampling is therefore very important. Strong wind also affects the deposition of airborne particles, which are deposited on the windward side of the snow dunes, as was observed in February 2021 on the Šijec bog.

As precipitation has a slightly acidic pH (5–6), some particles, such as carbonates, may dissolve shortly after they are deposited. This is particularly evident in rainwater, where daily temperature variations additionally affect the rate of particle dissolution. This is confirmed in the presented results, where pH 6.3 and 7.1 were measured in rainwater, indicating an increase after deposition in the sampler. In snow, this is not as apparent, although the process may occur during the melting of the snow sample, as indicating by the low pH results. The snowmelt water from 2020 had an average pH value of 6.3, while snowmelt from 2021 had an average pH value of 5.7. In 2020, we sampled older snow, which was exposed to the post-depositional processes longer than snow collected in 2021, which resulted in changes in physico-chemical parameters, hence a higher pH.

Rainwater pH can increase if it coincides with a dust event exceeding a value of 6.5 [38]. During dust events we only sampled snow, which did have higher values of pH (6.3), although they were the same for layers with and without records of a dust event. Considering average

pH values from the 2021 snowmelt, this indicates that the dust event does not have such a significant influence on pH of snowmelt as it does on rainwater.

Using multiple atmospheric deposition mediums provides a better understanding of processes that occur in the atmosphere and during deposition than interpretation based solely on passive atmospheric deposition analysis. Combining data from passive deposition and particles from precipitation prevents biased results, such as excluding certain groups of particles as they tend to be dissolved in precipitation.

### 4.2. Origin of the Atmospheric Particles and Relation to Dust Events

Samples from all atmospheric deposition mediums contained particles of both geogenic and anthropogenic origin. Most particles, including silicates, carbonates, organic matter, and partly Fe-oxyhydroxides, are of geogenic origin with local, regional, and distant sources. Similar groups of detrital material were found within peat in Spain [39]. They identified the main minerals as phyllosilicates, quartz, calcite, and feldspar and in lesser amount minerals from oxide, hydroxide, sulphide, sulphate, and chloride groups. It can be implied that certain minerals undergo changes after contact with the peat environment, however for a direct comparison, further research of mineral matter within the Šijec bog is needed. The Pokljuka plateau as the closest source, consists mainly of carbonate rocks, limestone, and dolomite. Besides being a source of carbonate particles, local geology is also an important source of carbonate dissolution products that represents the local source of silicate particles. The geology of Slovenia is diverse, though carbonates predominate. Therefore, geogenic regional sources provide mainly carbonate and silicate particles. Latter originating from carbonate dissolution and weathering of other lithological units [40].

The Pokljuka plateau is a densely forested area and is, therefore an important source of organic particles (plant remains, pollen, and leaves). The Šijec bog itself is a source of organic matter, as we find moss leaves and spores in the samples. Large (>10 μm) and rounded spores and pollen predominate among organic particles in all samples. The densely forested surroundings of the peatland indicate that local sources predominate, with other sources accounting for a minority of particles.

Iron bearing minerals are naturally present on the Pokljuka plateau [41], which may represent a small source of the geogenic Fe-oxyhydroxides.

Apart from local and regional sources, atmospheric particulate matter can originate from distant sources. Dust events transport particles over long distances. Particles deposited during dust events are predominantly of geogenic origin, although not limited to it.

Particles deposited during March 2020 dust event originated from Central Asia [29], whereas particles from the February 2021 dust event originated from North Africa [16]. The difference in origin is reflected in the properties and composition of the particles. Although silicate particles predominate in both events, the estimated percentage of their subgroups varies between events. In 2021, quartz was the predominating group, while in 2020, aluminosilicates were slightly more common (Table 3).

The size distribution of particles varies between the 2020 and 2021 dust events. In the 2020 Central Asia event, the increase in both, PM10 and PM2.5 values, is observed (Figure 2), whereas in 2021 North Africa event, only PM10 values are elevated (Figure 3). The size of the particles obtained during dust events also differs from average weather conditions. Generally, the particles during dust events are smaller. In 2020, the average size of aluminosilicate particles during and after the dust event are 11.0 μm (without including most 0.5 μm particles in calculation, see Section 3.2.3) and 14.1 μm, respectively. This is better observed in 2021, where aluminosilicates before and during the dust event on average measured 18.6 μm and 9.5 μm, respectively. Although smaller particles predominate in our dust event samples, larger particles are still capable of travelling over long distances, as described in Van der Does and co-authors [42]. They detected particles up to 450 μm in size from the Sahara Desert thousands of kilometers away from their origin. In March 2021, we sampled snow after the dust event, although no new snowfall occurred after the sampling in February 2021. Therefore, the March sample contained snow and particles deposited

prior to the February sampling, as well as newly deposited particles from the dust event. This affects the calculated average size of the particles as we cannot distinguish particles deposited during and before the dust event. Additionally, the average particle size in 2021 may be affected by large particles originating from Northern Italy [30] prior to the dust event in 2021.

Major local anthropogenic influences and sources include traffic, tourism, and other. These are more pronounced in winter (e.g., winter sports—biathlon, heating). Anthropogenic particles can also be transported over long distances [43]. The presence of Fe-oxyhydroxide spheres, shavings, and irregular shapes indicates anthropogenic origin [36], and may be associated with industry in the valleys surrounding the Pokljuka plateau.

During the years 2020 and 2021 there were periods of lockdowns in Slovenia due to the COVID-19 pandemic. This could affect the amount of particles present in the atmosphere, especially particles of anthropogenic origin. Although, we did not study the influence of lockdown in particular, its effect in peat chemical composition has been studied before [44]. They found out that even relatively short periods of lower air pollution are reflected in moss element contents [44].

Dust events represent a large mass of particulate matter and are therefore relatively easy to trace to their source area using satellite imaging. Since the period of deposition from dust event is known, it is possible to sample and separate the dust event particles from those originating from local and regional sources. In comparison, distinguishing particles based on origin during normal conditions would require other techniques to determine their chemical composition and trace the particles to their sources. However, using SEM/EDS we can assume their origin based on particle morphology and composition. Dust events can be detected within peat mineral matter due to increased inputs [23]. Even though the authors state there were visible differences between several detected dust events, it was not possible to determine their origin. Therefore, the study of recent events could provide possible origins for previous events of increased atmospheric deposition in peat.

## 5. Conclusions

The amount and composition of atmospheric deposition vary throughout the year and is highly dependent on weather conditions. Rare dust events represent periods of increased deposition and therefore greatly contribute to mineral matter input to peatlands. We assessed characteristics of the mineral matter input deposited on the Šijec bog on the Pokljuka plateau throughout the year. Multiple approaches (collection of rainwater, snow, and passive atmospheric deposition) were used to obtain solid particles which were characterized using SEM/EDS. This proved to be an efficient approach for determining the characteristics of the mineral matter in wet and dry deposition.

We obtained samples during average weather conditions and two dust events. The latter occurred in 2020 and 2021 and originated from Central Asia and North Africa, respectively. During average weather conditions, particles from local and regional sources predominate. Carbonates are present in all samples but are more abundant in passive deposition samples. They originate mostly from local and regional sources. The lesser amount of carbonates in rainwater and snow samples is probably due to carbonate dissolution in slightly acidic precipitation. In precipitation, sample carbonates were present as a more chemical-weathering resistant dolomite and as crystalline calcite or aragonite. The crystalline structure of carbonate particles indicates their crystallization on filters after they previously dissolved in the atmosphere or after deposition.

The most abundant particle group in all samples were silicates, most probably originating from carbonate dissolution residue or weathering of other rocks. Compared to carbonates, silicates are more resistant to chemical weathering. Additionally, they are a common weathering product (clay minerals). This contributes to their higher abundancy in the atmosphere. In the samples we divided silicates into quartz and aluminosilicate. Based on composition we further divided the latter into clay minerals with simple composition

(Si-Al-O) and other aluminosilicates, such as feldspar (Si-Al-(K,Na,Ca)-O) and olivine groups and their weathering products (Si-Al-(Mg,Fe)-O).

During dust events, silicate content increased, indicating that the dust consists mainly of silicate particles. In the March 2020 event, small (<5 μm), angular aluminosilicates predominate, which is consistent with the observed increase in PM10 and PM2.5 particles during the event period [28]. In contrast, in the February 2021 dust event the increase was only observed in PM10 levels [30] while PM2.5 levels were not as affected.

Important particle groups deposited on the Šijec bog are also organic particles (pollen, spores, plant remains, and leaves) and Fe-oxyhydroxides. The latter mainly originate from anthropogenic sources, which is indicated by their shapes, which do not occur naturally (spheres, irregular shape, agglomerates, and shavings).

The determined particle characteristics indicate that atmospheric deposition on the Pokljuka plateau is influenced by both geogenic and anthropogenic sources, of both local and long-distance origin. Although local origin may be of greater influence for certain particle groups (carbonates and organic matter), particles transported over long distances represent an important fraction of the annual atmospheric deposition.

**Supplementary Materials:** The following are available online at https://www.mdpi.com/article/10.3390/min12080982/s1, Figure S1. SEM field of view image with EDS elemental mapping results using 300× magnification of sample R2; Figure S2. SEM field of view image with EDS elemental mapping results using 600× magnification of sample M1-1; Figure S3. SEM field of view image with EDS elemental mapping results using 800× magnification of sample R1; Figure S4. EDS spectra of selected images in Figure 5 (particles from passive deposition); Figure S5. EDS spectra of selected images in Figures 7 and 8 (particles in snowmelt water).

**Author Contributions:** Conceptualization, V.P. and M.G. (Mateja Gosar); methodology, V.P., M.G. (Martin Gaberšek) and M.G. (Mateja Gosar); software, V.P.; validation, V.P. and M.G. (Mateja Gosar); formal analysis, V.P.; investigation, V.P. and M.G. (Mateja Gosar); resources, M.G. (Mateja Gosar); data curation, V.P.; writing—original draft preparation, V.P.; writing—review and editing, V.P., M.G. (Martin Gaberšek). and M.G. (Mateja Gosar); visualization, V.P. and M.G. (Mateja Gosar); supervision, M.G. (Mateja Gosar); project administration, M.G. (Mateja Gosar); funding acquisition, M.G. (Mateja Gosar) All authors have read and agreed to the published version of the manuscript.

**Funding:** This research was funded by the Slovenian Research Agency (ARRS) in the frame of the young researcher programme, in the frame of the research programme Groundwater and Geochemistry (P1-0020) and infrastructure programme "Geological information centre" (I0–0007 (A)). Financial assistance was also provided by the "Slovenian National Commission for UNESCO, National Committee of the International Geosciences and Geoparks Programme".

**Data Availability Statement:** Not applicable.

**Acknowledgments:** We would like to thank Miloš Miler for help during SEM/EDS analysis and field work. We also thank Slovenian Environment Agency (ARSO) for providing data on PM10 and PM2.5 values during dust events in 2020 and 2021.

**Conflicts of Interest:** The authors declare no conflict of interest.

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
