# Peer review of "Characterization of Atmospheric Deposition as the Only Mineral Matter Input to Ombrotrophic Bog"

_minerals, doi:10.3390/min12080982_

Round 1
Reviewer 1 Report
On Line 105 there is no indication on what height from ground were placed the stubs.
On my personal opinion although the effort could be enormous, the amount of particles potentially captured from exposed stubs are few compared to the volume of water and snow captured.
In 2020 in some world region in the analisys period there was lockdown, have you evaluated the infuence of that ? ( pollution caused by industrial and transport could be lower than expected)
Could be interesting continue the investigation on a more extended period of time to extend the statistical analisys.
Author Response
Thank you for your comments and suggestions. Below we present answers to your individual comments, while in the manuscript, the corrections are highlighted in yellow.
- On Line 105 there is no indication on what height from ground were placed the stubs.
We added the height information (180 cm).
- On my personal opinion although the effort could be enormous, the amount of particles potentially captured from exposed stubs are few compared to the volume of water and snow captured.
Thank you for your comment. As you mentioned, we are not able to perform dust characterization solely based on the passive deposition, which is why we complement it with other sampling mediums. However, passive deposition is a good indicator of what kind of particles are deposited on a specific surface in a certain period of time. We specified that the amount is much smaller compared to other mediums and emphasised the importance of passive deposition in lines 247–249.
- In 2020 in some world region in the analysis period there was lockdown, have you evaluated the influence of that? (pollution caused by industrial and transport could be lower than expected).
Thank you, we agree with your comment. In our paper, however, we did not consider it as we do not have a lockdown or pre-lockdown representative sample. In March 2020 when lockdown in Slovenia started, we sampled snow, which included particles before and after the lockdown, while in later samples, the lockdown was less severe (industry operated as normal, while tourism was limited). We agree that it would be interesting to study the influence lockdown had on atmospheric deposition. In the Pokljuka area there was another study by other researcher who investigated the lockdown influence on surface peat moss. We added a comment on the influence of lockdown and cited the mentioned paper (lines 535–537).
- Could be interesting continue the investigation on a more extended period of time to extend the statistical analysis.
We agree with your comment and hope to extend it in the future. As mentioned in previous comment it would also be interesting to extend the study, to obtain representative sample of post-lockdown or other unusual events, such as the lockdown.
Reviewer 2 Report
The manuscript minerals-1783866 “SEM/EDS characterization of atmospheric deposition as the only mineral matter input to ombrotrophic bog” is a fundamental study of the mineral matter in peat bog. The undoubted merit of the authors is the routine and detailed analysis of mineral microparticles based on the SEM-EDS analysis. However, there are a few areas where I think the authors can strengthen the manuscript. These are not related to the science largely, but the presentation of their work in the text and the clarity of the findings they are trying to convey. I view these changes as moderate as they will not require substantial revision, but careful tweaking and extending of a few sections of the manuscript. I think that the manuscript would be a valuable contribution to the Minerals.
Major comments.
1. Main conclusions in sections 3.3 and 3.4 should be more discussed with exemplifying similar studies from other parts of the world. In other words, the authors should justify their conclusions through comparing their results with others’ works, with more emphasis on the similarities and differences of their findings with those works. In general, this is a problem which is present throughout the discussion chapter. The authors should give more credit to others’ works and discuss their findings in the context of previous work. A moderate revision to the discussion chapter is thus recommended.
2. Removing the method of determination from the title is recommended. The content of the article is about minerals in bog sediments. The content should not change depending on the analysis method. The emphasis on the methodology and results on the results of SEM will allow readers to understand that electron microscopy is one of the only methods for determining minor and rare mineral inclusions in the peat. For example, “Characterization of atmospheric deposition as the only mineral matter input to ombrotrophic bog”.
3. A review of possible processes of transformation of mineral particles of atmospheric precipitation and traces of in situ modifications in the studied sediments will expand the significance of the studies of the Šijec bog.
4. Divide the Results and Discussion sections.
5. “EDS elemental mapping” should be shown in a separate figure or supplementary materials.
6. To identify minerals, it is necessary to describe their chemical composition according to EDS analysis data. Therefore, EDS spectra should be added to some images.
Minor comments:
7. Line 27-28. Please study the following papers and expand the review of the mineral importance in bog sediments and reference.
Shotyk, W., 1988. Review of the inorganic geochemistry of peats and peatland waters. Earth-Science Reviews 25, 95–176. https://doi.org/10.1016/0012-8252(88)90067-0
López-Buendía, A.M., Whateley, M.K.G., Bastida, J., Urquiola, M.M., 2007. Origins of mineral matter in peat marsh and peat bog deposits, Spain. International Journal of Coal Geology 71, 246–262. https://doi.org/10.1016/j.coal.2006.09.001
Smieja-Król, B., FiaÅ‚kiewicz-KozieÅ‚, B., 2014. Quantitative determination of minerals and anthropogenic particles in some Polish peat occurrences using a novel SEM point-counting method. Environmental Monitoring and Assessment 186, 2573–2587. https://doi.org/10.1007/s10661-013-3561-0
Sjöström, J.K., Martínez Cortizas, A., Hansson, S. V., Silva Sánchez, N., Bindler, R., Rydberg, J., Mörth, C.-M.M., Ryberg, E.E.S., Kylander, M.E., 2020. Paleodust deposition and peat accumulation rates – Bog size matters. Chemical Geology 554, 119795. https://doi.org/10.1016/j.chemgeo.2020.119795
Awid-Pascual, R., Kamenetsky, V.S., Goemann, K., Allen, N., Noble, T.L., Lottermoser, B.G., Rodemann, T., 2015. The evolution of authigenic Zn–Pb–Fe-bearing phases in the Grieves Siding peat, western Tasmania. Contributions to Mineralogy and Petrology 170, 17. https://doi.org/10.1007/s00410-015-1167-y
Kylander, M.E., Martínez-Cortizas, A., Bindler, R., Greenwood, S.L., Mörth, C.-M., Rauch, S., 2016. Potentials and problems of building detailed dust records using peat archives: An example from Store Mosse (the “Great Bog”), Sweden. Geochimica et Cosmochimica Acta 190, 156–174. https://doi.org/10.1016/j.gca.2016.06.028
Rudmin, M., Ruban, A., Savichev, O., Mazurov, A., Dauletova, A., Savinova, O., 2018. Authigenic and Detrital Minerals in Peat Environment of Vasyugan Swamp, Western Siberia. Minerals 8, 1–13. https://doi.org/10.3390/MIN8110500
Syrovetnik, K., Puura, E., Neretnieks, I., 2004. Accumulation of heavy metals in Oostriku peat bog, Estonia: Site description, conceptual modelling and geochemical modelling of the source of the metals. Environmental Geology 45, 731–740. https://doi.org/10.1007/s00254-003-0931-x
Steinmann, P., Shotyk, W., 1997. Geochemistry, mineralogy, and geochemical mass balance on major elements in two peat bog profiles (Jura Mountains, Switzerland). Chemical Geology 138, 25–53. https://doi.org/10.1016/S0009-2541(96)00171-4
8. Line 69. Decipher the abbreviation “PM” at the first mention in the text.
9. Line 189. Specify the “spot sizes” dimension.
10. Line 201. Specify image resolution or view fields for corresponding magnifications.
Author Response
The manuscript minerals-1783866 “SEM/EDS characterization of atmospheric deposition as the only mineral matter input to ombrotrophic bog” is a fundamental study of the mineral matter in peat bog. The undoubted merit of the authors is the routine and detailed analysis of mineral microparticles based on the SEM-EDS analysis. However, there are a few areas where I think the authors can strengthen the manuscript. These are not related to the science largely, but the presentation of their work in the text and the clarity of the findings they are trying to convey. I view these changes as moderate as they will not require substantial revision, but careful tweaking and extending of a few sections of the manuscript. I think that the manuscript would be a valuable contribution to the Minerals.
Thank you for your good opinion and all suggestions for improvement of our manuscript. We comment all suggestions bellow, while corrections in the manuscript are highlighted in green colour.
- Main conclusions in sections 3.3 and 3.4 should be more discussed with exemplifying similar studies from other parts of the world. In other words, the authors should justify their conclusions through comparing their results with others’ works, with more emphasis on the similarities and differences of their findings with those works. In general, this is a problem which is present throughout the discussion chapter. The authors should give more credit to others’ works and discuss their findings in the context of previous work. A moderate revision to the discussion chapter is thus recommended.
We expanded discussion, where we focused on the comparison with similar previous research, particularly in the section 4.2 (previously 3.4). The added changes are labelled with green colour.
- Removing the method of determination from the title is recommended. The content of the article is about minerals in bog sediments. The content should not change depending on the analysis method. The emphasis on the methodology and results on the results of SEM will allow readers to understand that electron microscopy is one of the only methods for determining minor and rare mineral inclusions in the peat. For example, “Characterization of atmospheric deposition as the only mineral matter input to ombrotrophic bog”.
Thank you for the suggestion. We changed the title as the method is already mentioned in keywords.
- A review of possible processes of transformation of mineral particles of atmospheric precipitation and traces of in situ modifications in the studied sediments will expand the significance of the studies of the Šijec bog.
Thank you for the comment. We agree, and this is actually an important part of our study, that we do not mention in this paper. We are currently in the process of studying chemical composition of water and peat, as well as solid particles in both. Results of this and current studies will be used to assess the processes and changes within the peat and how it reflects in peat mineral matter itself.
- Divide the Results and Discussion sections.
Thank you, we separated the sections, with description of individual samples labelled as results, and last two sections labelled as discussion.
- “EDS elemental mapping” should be shown in a separate figure or supplementary materials.
As supplementary material we added figures of fields for elemental mapping and the results of such mapping.
- To identify minerals, it is necessary to describe their chemical composition according to EDS analysis data. Therefore, EDS spectra should be added to some images.
In supplementary material we also added spectra of the existing figures and added a reference.
- Line 27-28. Please study the following papers and expand the review of the mineral importance in bog sediments and reference.
Thank you for suggestions, we included some of them as well as other references to extend the description of mineral matter research in peatlands, and made a better connection to the present atmospheric deposition. Included in lines 30–40.
- Line 69. Decipher the abbreviation “PM” at the first mention in the text.
Thank you for the comment. We explained the abbreviation PM the first time we mention particulate matter. We also added explanation for both abbreviations, PM10 and PM2.5.
- Line 189. Specify the “spot sizes” dimension.
We corrected the range of used spot sizes to 48–50, as it is also indicated in the figures. Our JEOL electronic microscope manual only gives information about spot size without units and does not specify the actual diameter of the electron beam.
- Line 201. Specify image resolution or view fields for corresponding magnifications
Thank you, we added the reference to the supplementary material where we provided figures of elemental mapping. On the mapped fields, the image resolution is provided.
Reviewer 3 Report
In the submitted manuscript (SEM/EDS characterization of atmospheric deposition as the only mineral matter input to ombrotrophic bog) by Valentina Pezdir and associates, the investigators have used traditional method to examine the mineral matter deposited on Šijec bog. The study presents a major problem in that is almost purely descriptive and lacks a hypothesis. In addition, there is no statistical analysis in the paper, which lacks credibility.
Author Response
In the submitted manuscript (SEM/EDS characterization of atmospheric deposition as the only mineral matter input to ombrotrophic bog) by Valentina Pezdir and associates, the investigators have used traditional method to examine the mineral matter deposited on Šijec bog. The study presents a major problem in that is almost purely descriptive and lacks a hypothesis. In addition, there is no statistical analysis in the paper, which lacks credibility.
Thank you for your comments. We made corrections to the English language. We highlighted the corrections in the manuscript in blue colour.
- The study presents a major problem in that is almost purely descriptive and lacks a hypothesis.
We added more descriptions of the importance of mineral matter in peat and its connection to present atmospheric deposition to emphasise the main goal of our research. The goal of the study is to chemically and morphologically describe particles deposited on the peatland, as well as to determine the main possible sources and importance of other factors contributing to the mineral matter – such as dust events. Description of importance of mineral matter in peat is provided in lines 30–40 and emphasis on the goal is presented and updated in lines 72–79.
- In addition, there is no statistical analysis in the paper, which lacks credibility.
Thank you for the comment about lack of statistical analysis. We acknowledge that we did not perform the statistical analysis as we more focused on description of particle properties which are harder to statistically evaluate, e.g., shape, morphology, mineralogy, chemical composition, and origin (geogenic/anthropogenic). Mineral composition assessment is based only on elemental mapping and is not suitable for statistics. Additionally, the chemical composition is based on semi-quantitative analysis, therefore, statistical analysis would be unreliable. We also have a relatively small number of samples, which further makes it difficult to perform a good statistical analysis. We will later use the presented results and extend the research with results of water chemical composition in the peatland surroundings, chemical composition of peat mineral and organic matter and results of solid particles within waters and peat.